# Clinical Predictors of Early Trial Discontinuation for Patients Participating in Phase I Clinical Trials in Oncology

**DOI:** 10.3390/cancers13102304

**Published:** 2021-05-11

**Authors:** Joeri A. J. Douma, Laurien M. Buffart, Ramy Sedhom, Mariette Labots, Willemien C. Menke-van der Houven van Oordt, Mikkjal Skardhamar, Anthony De Felice, Esther Lee, Divya Dharmaraj, Nilofer S. Azad, Michael A. Carducci, Henk M. W. Verheul

**Affiliations:** 1Cancer Center Amsterdam, Amsterdam University Medical Center, Department of Medical Oncology, Vrije Universiteit Amsterdam, 1081 HV Amsterdam, The Netherlands; j.douma@nki.nl (J.A.J.D.); m.labots@amsterdamumc.nl (M.L.); c.menke@amsterdamumc.nl (W.C.M.-v.d.H.v.O.); m.skardhamar@amsterdamumc.nl (M.S.); 2Department of Physiology, Radboudumc, 6525 GA Nijmegen, The Netherlands; Laurien.buffart@radboudumc.nl; 3Sidney Kimmel Comprehensive Cancer Center at Johns Hopkins, Baltimore, MD 21231, USA; rsedhom1@jhmi.edu (R.S.); adefeli3@jhu.edu (A.D.F.); elee138@jhu.edu (E.L.); ddharma1@jhu.edu (D.D.); Nazad2@jhmi.edu (N.S.A.); Carducci@jhmi.edu (M.A.C.); 4Department of Medical Oncology, Radboudumc, 6525 GA Nijmegen, The Netherlands

**Keywords:** phase I trial, drug development, early trial discontinuation, clinical predictors, hyponatremia

## Abstract

**Simple Summary:**

About 20% of patients with cancer who participate in a phase I clinical trial discontinue the trial early. Early trial discontinuation is undesirable both for the wellbeing of the patient as well as for the trial efficiency and the development of new anticancer drugs. We investigated which clinical predictors at baseline were significantly associated with early trial discontinuation of patients with cancer who participated in phase I clinical trials. The clinical predictors which were identified in this study were hyponatremia, elevated alkaline phosphatase level, performance score of 1 or higher and opioid use. Hyponatremia especially, which was the strongest predictor, should be considered to be used as an additional eligibility criterion in order to reduce the early trial discontinuation of patients with cancer who participate in phase I clinical trials.

**Abstract:**

Despite stringent eligibility criteria for trial participation, early discontinuation often occurs in phase I trials. To better identify patients unlikely to benefit from phase I trials, we investigated predictors for early trial discontinuation. Data from 415 patients with solid tumors who participated in 66 trials were pooled for the current analysis. Early trial discontinuation was defined as (i) trial discontinuation within 28 days after start of treatment or (ii) discontinuation before administration of the first dosage in eligible patients. Multilevel logistic regression analyses were conducted to identify predictors for early trial discontinuation. Eighty-two participants (20%) demonstrated early trial discontinuation. Baseline sodium level below the lower limit of normal (OR = 2.95, 95%CI = 1.27–6.84), elevated alkaline phosphatase level > 2.5 times the upper limit of normal (OR = 2.72, 95%CI = 1.49–4.99), performance score ≥ 1 (OR = 2.07, 95%CI = 1.03–4.19) and opioid use (OR = 1.82, 95%CI = 1.07–3.08) were independent predictors for early trial discontinuation. Almost 50% of the patients with hyponatremia and all four patients in whom all four predictors were present together discontinued the trial early. Hyponatremia, elevated alkaline phosphatase level, performance score ≥ 1 and opioid use were identified as significant predictors for early trial discontinuation. Hyponatremia was the strongest predictor and deserves consideration for inclusion in eligibility criteria for future trials.

## 1. Introduction

Phase I trials in oncology are essential for drug development and evaluation of novel treatment strategies [1]. Traditionally, phase I trials are designed to test the safety and side effects of a new drug or treatment strategy, with the aim of establishing the recommended dose for subsequent trials [2]. Since patients who are referred for phase I trial participation are usually heavily pretreated, the eligibility criteria for these trials are stringent to minimize the potential harm of study treatment. To be eligible for participation for most trials, patients must have an adequate performance status (PS) and organ function and a minimal life expectancy of 3 months without treatment [1,2,3]. As a result of these strict criteria, only 30% of referred patients are eligible, with an insufficient PS being the main reason for ineligibility [4,5]. 

Despite the stringent selection, previous studies revealed that 15% of participants in phase I trials discontinue within three weeks [6]. Patients who discontinue the trial early are not likely to benefit from study treatment. Moreover, participation in a phase I trial may yield additional burdens for patients due to possible adverse effects, multiple (long) visits at the clinic, invasive procedures and sometimes even extra financial costs [7,8,9]. In addition, study outcomes are negatively influenced by high rates of patients discontinuing the trial early. Approximately 70% of early trial discontinuation is due to a deteriorating condition caused by progressive disease and/or concomitant medical events that are not related to the study treatment [6]. Furthermore, patients who discontinue trial participation during the evaluation period because of non-drug-related events need to be replaced for evaluation of safety and dose limiting toxicity (DLT). The higher accrual necessary for adequate evaluation is detrimental to the efficient conduct of trials [5,6]. 

Prevention of early dropout may benefit the conduct and outcome of a trial [2]. In two previous studies, predictors for early trial discontinuation in participants of phase I trials conducted before 2010 have been reported [6,10]. The results of these studies were inconsistent, with the exception of PS and serum alkaline phosphatase, which were consistently identified as predictors. In addition, these studies are relatively old and did not include newer treatment strategies such as vaccines, radiopharmaceuticals and immunotherapy. We here investigated clinical or patient-related predictors of early trial discontinuation of participants of various types of phase I trials, which could be used as additional eligibility criteria across trials in the future. Furthermore, we studied the rate of 90-day mortality, which is a frequently used eligibility criterion in phase I trials [3], both in patients with and without early trial discontinuation to emphasize that early trial discontinuation is a relevant measure. 

## 2. Materials and Methods

### 2.1. Study Design and Population

For the current retrospective analysis, we pooled data from patients with solid tumors who participated in phase I trials at the Amsterdam UMC, Vrije Universiteit Amsterdam, in Amsterdam, The Netherlands, and at the Sidney Kimmel Comprehensive Cancer Center at Johns Hopkins in Baltimore, US. The phase I trials were conducted between January 2013 and January 2019. All data were collected in the context of the phase I trial and were retrieved from the electronic health records of the hospital. All study protocols were approved by Institutional Review Boards prior to patient recruitment and conducted in accordance with the International Conference on Harmonization E6 Guidelines for Good Clinical Practice. All patients provided written informed consent before participation in the phase I trials. 

### 2.2. Outcome

Early trial discontinuation was defined as (i) trial discontinuation within 28 days of administration of the first dosage of the investigated drug or (ii) discontinuation before administration of the first dosage in patients who were found to be eligible after the screening process. Patients who met these criteria would not have had a chance to benefit from the treatment, although they may have experienced toxicity or complications from treatment or investigations during the screening process. Ninety-day mortality was defined as (i) death from any cause within 90 days of administration of the first dosage of the investigated drug or (ii) death from any cause before administration of the first dosage in patients who were found to be eligible after screening. 

### 2.3. Potential Predictors

Potential predictors and cut-offs were defined before reviewing the data and selected based on their previously reported association with early trial discontinuation, 90-day mortality, overall survival or clinical relevance in participants of phase I oncology trials [6,10,11]. The following baseline characteristics were selected: opioid use, number of metastatic sites [6], body mass index (BMI), ECOG/WHO PS [6,10], history of thromboembolism [12], hemoglobin [13], platelet count [12], white blood cell count [6], lymphocytes [6], absolute neutrophil count, neutrophils-to-lymphocytes (NTL) ratio [14], serum sodium [15,16], creatinine clearance [10], serum albumin [10], serum alkaline phosphatase [10], serum aspartate aminotransferase [10], serum alanine aminotransferase [10], serum lactate dehydrogenase [6] and the Charlson Comorbidity Index [17]. BMI was calculated from measurements of body height and weight (body weight/height^2^, kg/m^2^). Creatinine clearance was estimated by using the Chronic Kidney Disease Epidemiology Collaboration formula [18]. The NTL ratio was calculated by dividing the neutrophils by the lymphocytes. All of the above predictors were dichotomized according to the cut-off values that were found in the literature to facilitate interpretability and use in clinical practice (Table 1).

### 2.4. Statistical Analysis

Univariable and multivariable logistic regression analyses were conducted to identify factors that were significantly associated with early trial discontinuation. Potentially relevant variables identified from the univariable analysis were checked for multicolinearity (r ≥ 0.60). A stepwise forward selection procedure was used to build the multivariable regression model, starting with the variable that was most strongly associated with the outcome in the univariable regression model. Subsequently, the next strongest variable was selected after controlling for the first variable. This procedure was repeated until no variables with an association with the outcome at a significance level of *p* < 0.05 could be added to the model. A random intercept was added to the model to take into account the clustering of patients within studies. We also checked whether the hospital where the trial was performed (Amsterdam UMC, Vrije Universiteit Amsterdam versus the Sidney Kimmel Comprehensive Cancer Center at Johns Hopkins), the type of therapy (immunotherapy versus other types of treatment) or the type of the trial (phase I versus phase I/II) were non-patient-related predictors of early trial discontinuation, but none of the associations were statistically significant. As one study (NCT02058901) provided most of the data, we performed sensitivity analyses on the data excluding that study. Descriptive statistics and a chi-squared test were used to compare the rate of 90-day mortality both in patients with and without early trial discontinuation. The odds ratio (OR) and 95% confidence interval (CI) of the models were reported. Crosstabs were generated to present the proportion of patients with early trial discontinuation and separately for all the significant predictors identified in the multivariable analyses. Subsequently, we calculated the positive predictive value for the combination of predictors. Analyses were conducted with SPSS version 22 and RStudio version 3.4.2.

## 3. Results

Data from 415 patients recruited from 66 phase I trials were analyzed. In 21% of the trials, immunotherapy alone was investigated, in 20% immunotherapy in combination with targeted therapy, in 18% targeted therapy alone, in 16% targeted therapy in combination with cytotoxic therapy and in the remaining 25% other types of drugs or combinations of drugs were investigated (Appendix A). The mean age of the patients was 61 (standard deviation (SD) = 11) years, half of patients were women and 69% had a PS of 1 (Table 1). In total, 82 patients (20%) met the criteria for early trial discontinuation. Five patients (5/82 or 6%) discontinued trial participation before administration of the first dosage. For the patients who received the first administration of the drug and discontinued the trial early, the average time to trial discontinuation was 17 days (SD = 7). Thirty-six patients (44%) discontinued the trial early due to progressive disease, 15 (18%) due to physical deterioration, 10 patients (12%) due to toxicity that was not dose limiting and 10 patients (12%) discontinued the trial upon their own request (Table 2). 

In the multivariable model, a reduced serum sodium level (OR = 2.95, 95%CI = 1.27–6.84), an elevated alkaline phosphatase level above 2.5 times the upper limit of normal (ULN) (OR = 2.72, 95%CI = 1.49–4.99), a PS of ≥1 (OR = 2.07, 95%CI = 1.03–4.19) and opioid use (OR = 1.82, 95%CI = 1.07–3.08) were significantly and independently associated with early trial discontinuation (Table 3). The positive predictive value for early trial discontinuation was 46% for hyponatremia, 39% for elevated alkaline phosphatase level, 38% for opioid use and 31% for PS ≥ 1. Hyponatremia was the predictor with the highest positive predictive value, based on the fact that 13 (46%) of the 28 patients with hyponatremia discontinued the trial early. The median (interquartile range) sodium level was 139 (137–141) mmol/L, with a lower limit of normal of 135 mmol/L in both hospitals. Furthermore, all four predictors were present in four patients. The positive predictive value in this situation was 100%. Sensitivity analyses on the data excluding the trial with the most participants yielded comparable results.

In total, 59% of the patients who discontinued early died within 90 days of starting the trial compared to 13% of the patients who did not discontinue the trial early (*p* < 0.001). 

## 4. Discussion

Early trial discontinuation in phase I trials is an important and common problem, for which a rate of 20% was found in this study. Hyponatremia, PS ≥ 1, opioid use and elevated alkaline phosphatase level were identified as significant independent predictors for early trial discontinuation amongst patients who participated in phase I trials. Furthermore, the rate of 90-day mortality was significantly higher in patients who discontinued the trial early than in patients who did not.

Importantly, one novel finding is that hyponatremia was identified as a significant predictor for early trial discontinuation. In earlier studies, hyponatremia has not been identified or investigated as a potential predictor for early trial discontinuation [6,10] but was has been to be predictive for 90-day mortality and overall survival in patients participating in phase I oncology trials [15,16] and for overall survival in patients with breast, colorectal and lung cancer [19]. In an ancillary analysis, we found that hyponatremia was also significantly associated with 90-day mortality (OR = 3.62, 95%CI = 1.63–8.02). None of the included phase I trials in this study, nor any of the cancer trials examined in a review [3], used serum sodium level as eligibility criterion. There are several possible explanations why hyponatremia could be predictive for early trial discontinuation. Hyponatremia can be caused by the syndrome of inappropriate anti-diuretic hormone (SIADH) (due to cancer, pain or co-medication), by hypo- or hypervolemia and, although rare, by reduced salt intake [19]. We could not identify any corresponding clinical etiology (e.g., comorbidity, use of painkillers, co-medication) for the hyponatremia in the patients from our trials. Unfortunately, extensive diagnostics (e.g., MRI scan of the brain) to determine the cause of the hyponatremia were not performed in most patients. However, hyponatremia may also reflect a more advanced, otherwise undetected, stage of cancer in general. Further exploration of the etiology and the optimal cut-off value of hyponatremia should be considered in future studies and, more importantly, the role of hyponatremia as an eligibility criterion should be further investigated and might be used in future phase I trials. In the current study, lowering the cut-off value to the threshold of grade 1 hyponatremia (sodium < 130 mmol/L), conforming to the Common Terminology Criteria for Adverse Events (CTCAE), did not result in a clinically relevant improvement in the positive predictive value.

The results of the current study also indicated that opioid use was predictive for early trial discontinuation. The use of opioids has not been investigated earlier as a predictor for early trial discontinuation or mortality in participants in phase I trials. However, opioid use is an expression of cancer-related pain, which was found to be a significant predictor of overall survival in a systematic review and in meta-analysis in patients with different types of cancer [20,21]. Both PS and an elevated alkaline phosphatase level were identified as significant predictors of early trial discontinuation. This is in line with two earlier studies, in which elevated alkaline phosphatase level and PS were identified as significant predictors for early trial discontinuation in phase I trial participants [6,10]. 

Almost 60% of the patients with early trial discontinuation died within 90 days. Both early trial discontinuation and 90-day mortality are most likely caused by early progression, study-related toxicities, other complications or true inefficacy of the treatment in combination with an incorrectly estimated life expectancy. Patients who will die within 90 days should not be included in a trial [3] and should be offered advanced care planning, aggressive symptom management and discussion of end of life [22,23]. A better selection of patients to be included in a trial might reduce both early trial discontinuation as well as 90-day mortality. 

A possible limitation of this study might have been the inclusion of heterogeneous trials that investigated newer therapies like vaccines, kinase inhibitors and radiopharmaceuticals, as well as cytotoxic agents. On the other hand, the heterogeneity may enhance the generalizability of the results to other phase I trials. Furthermore, the current study, with a sample size of 415 and 82 events, is one of the largest studies evaluating predictors of early trial discontinuation and made it possible to study up to 8 predictors in the multivariable analysis. 

In general, phase I trial participation offers intensified care for cancer-related symptoms, which can improve quality of life and may even improve survival [24,25,26], regardless of the phase I study drugs administered. This study identified two new predictors for early trial discontinuation in patients with cancer participating in phase I trials. Notably, a high positive predictive value of the predictor is important in this setting because participation could be harmful to patients, but unnecessary exclusion for a potential beneficial drug is also undesirable. The positive predictive value of hyponatremia is quite high and has clinical relevance, but sodium level is currently not used as an eligibility criterion. Previous studies have shown that only 15% of patients experience a clinical benefit from experimental treatment in phase 1 trials at 6 months [27,28]. This proportion is likely substantially lower in patients with early trial discontinuation. Due to the low chance of clinical benefit in combination with the potential harm of participation in a phase I clinical trial, a positive predictive value of around 50% for early trial discontinuation supports the use of hyponatremia as an exclusion criterion and to exclude these patients from participation in phase I trials. 

The results of this study indicate that hyponatremia might be useful for trial selection and we propose that this criterion deserves further investigation. Changes to the conduct of early phase trials should be carefully considered because they potentially have far-reaching effects [29]. Unnecessary restrictive eligibility criteria should be avoided to maintain generalizability to the patients who will ultimately be treated with the investigated drugs [30]. More stringent eligibility criteria are warranted if research indicates that patients have a high risk for an adverse outcome when participating in the study, for example early trial discontinuation. On the other hand, some eligibility criteria (like brain metastases, HIV infection and concurrent malignancies) that are not likely to protect the safety of trial participants should be loosened or abandoned [30]. As patients with hyponatremia have a high risk of early trial discontinuation, we hypothesize that adjustment of the eligibility criteria with the addition of hyponatremia might contribute to a better selection of patients. Therefore, prospective validation of our results is warranted to determine whether adjustments of eligibility criteria are needed and to investigate whether the threshold for hyponatremia could be lowered. In the meanwhile, clinicians should be alerted that patients with hyponatremia, opioid use, PS ≥ 1 and an elevated alkaline phosphatase level are at risk for an unfavorable outcome and should be closely monitored, with appropriate initiation of supportive or palliative care if needed.

## 5. Conclusions

In conclusion, hyponatremia, opioid use, PS ≥ 1 and an elevated alkaline phosphatase level were identified as predictors for early trial discontinuation amongst patients with cancer participating in phase I trials. Hyponatremia was the strongest predictor with a positive predictive value of 46% and discussion should be centered on this as a possible exclusion criterion in future phase I trials in oncology. Furthermore, the rate of 90-day mortality was significantly higher in patients who discontinued the trial early than in patients who did not. 

## Figures and Tables

**Table 1 cancers-13-02304-t001:** Baseline patient characteristics (*n* = 415).

Characteristic	Statistics
Age, mean (SD) years	61 (11)
Sex, *n* (%) women	204 (49)
Recruited in the Netherlands, *n* (%)	154 (37)
Primary tumor, *n* (%)	
Gastrointestinal	207 (50)
Genitourinary	42 (10)
Lung cancer	32 (8)
Skin and soft tissue cancer	31 (7)
Breast cancer	28 (7)
Gynecological	19 (5)
Head and neck	18 (4)
Glioblastoma	17 (4)
Neuroendocrine carcinoma	8 (2)
Others	13 (3)
ECOG/WHO performance status, *n* (%) ^a^	
0	118 (29)
1	285 (69)
>1	5 (1)
BMI, mean (SD) (kg/m^2^)	26 (5)
BMI < 18.5, *n* (%) ^b^	20 (5)
Opioid use, *n* (%) ^c^	177 (43)
Three or more metastatic sites, *n* (%)	118 (77)
Any comorbidity other than primary malignancy, *n* (%)	104 (25)
Diabetes mellitus, *n*	50
COPD, *n*	20
Myocardial infarction, *n*	12
Cerebrovascular accident or transient ischemic attack, *n*	11
Peripheral vascular disease, *n*	7
Liver disease, *n*	7
Peptic ulcer disease, *n*	7
Kidney disease, *n*	7
Connective tissue disease, *n*	6
Heart failure, *n*	2
History of thromboembolism, *n* (%) ^d^	57 (14)
Laboratory tests	
Hemoglobin (mmol/L) < 7.45, *n* (%)	221 (53)
White blood cell count (10^9^/L) > ULN, *n* (%)	62 (15)
Lymphocytes (10^9^/L) < LLN, *n* (%) ^e^	70 (17)
Neutrophils (10^9^/L) > ULN, *n* (%) ^f^	56 (14)
Neutrophils-to-lymphocytes (NTL) ratio > 5, *n* (%) ^g^	154 (38)
Platelets (10^9^/L) > 440, *n* (%) ^h^	30 (7)
Sodium (mmol/l) < LLN, *n* (%) ^i^	28 (7)
Creatinine clearance (ml/min/1.73 m^2^) < 60, *n* (%) ^j^	56 (14)
Albumin (g/L) < 35, *n* (%) ^k^	125 (30)
Lactate dehydrogenase (U/L) > 600, *n* (%) ^l^	34 (11)
AST (U/L) > ULN, *n* (%) ^m^	132 (32)
ALT (U/L) > ULN, *n* (%) *^n^*	59 (14)
Alkaline phosphatase (U/L) > 2.5 x ULN, *n* (%) ^o^	67 (16)

a = *n* − 7, b = *n* − 2, c = *n* − 2, d = *n* − 1, e = *n* − 3, f = *n* − 3, g = *n* − 5, h = *n* − 1, i = *n* − 2, j = *n* − 3, k = *n* − 3, l = *n* − 107, m = *n* − 8, o = *n* − 2, *p* = *n* − 2, SD = standard deviation, *n* = number of patients, ECOG/WHO = Eastern Cooperative Oncology Group/World Health Organization, BMI = body mass index, COPD = chronic obstructive pulmonary disease, ULN = upper limit of normal, LLN = lower limit of normal, AST = aspartate aminotransferase, ALT = alanine aminotransferase.

**Table 2 cancers-13-02304-t002:** Reasons for early trial discontinuation.

Characteristic	*n* (%)
Early trial discontinuation	82 (20)
Reasons for early trial discontinuation (*n* = 82) ^a^	
Progressive disease	36 (44)
Physical deterioration (not otherwise specified)	15 (18)
Patient’s request	10 (12)
Toxicity (not dose limiting)	10 (12)
Dose-limiting toxicity or serious adverse event	5 (6)
Death	4 (5)
Protocol violation	1 (1)
90-day mortality ^b^	88 (22)

a = *n* − 1, b = *n* − 19.

**Table 3 cancers-13-02304-t003:** Results of the univariable, multivariable and logistic regression analyses for early trial discontinuation.

Predictor	Univariable Analysis		Multivariable Analysis	
	OR (95%CI)	*p*	OR (95%CI)	*p*
ECOG/WHO performance status				
1 vs. 0	**2.98 (1.51–5.86)**	**<0.01**	**2.07 (1.03–4.19)**	**0.04**
BMI (kg/m^2^)	0.96 (0.92–1.01)	0.14		
<18.5 vs. ≥18.5	**2.92 (1.15–7.41)**	**0.02**		
Use of opioids vs. non-use	**2.44 (1.48–4.01)**	**<0.01**	**1.82 (1.07–3.08)**	**0.03**
Metastatic sites				
≥3 metastatic sites vs. <3	1.45 (0.89–2.35)	0.14		
Charlson Comorbidity Score				
≥1 vs. 0	0.74 (0.41–1.33)	0.31		
History of thromboembolism				
Yes vs. no	1.23 (0.63–2.42)	0.54		
Laboratory tests				
Hemoglobin (mmol/L) < 7.45	1.58 (0.96–2.59)	0.07		
White blood cell count (10^9^/L) > ULN	1.67 (0.90–3.11)	0.10		
Lymphocytes (10^9^/L) < LLN	1.12 (0.60–2.10)	0.73		
Neutrophils (10^9^/L) > ULN	**2.20 (1.18–4.11)**	**0.01**		
Neutrophils-to-lymphocytes (NTL) ratio > 5	1.52 (0.93–2.49)	0.09		
Platelets (10^9^/L) > 440	**2.55 (1.16–5.60)**	**0.02**		
Sodium (mmol/L) < LLN	**4.04 (1.84–8.88)**	**<0.01**	**2.95 (1.27–6.84)**	**0.01**
Creatinine clearance (mL/min/1.73 m^2^) < 60	1.00 (0.49–2.03)	1.00		
Albumin (g/L) < 35	**2.56 (1.55–4.23)**	**<0.01**		
Lactate dehydrogenase (U/L) > 600	**2.25 (1.03–4.93)**	**0.04**		
AST (U/L) > ULN	**2.22 (1.34–3.68)**	**<0.01**		
ALT (U/L) > ULN	1.33 (0.69–2.57)	0.39		
Alkaline phosphatase (U/L) > 2.5 × ULN	**3.36 (1.90–5.93)**	**<0.01**	**2.72 (1.49–4.99)**	**0.001**

OR = odds ratio, CI = confidence interval, ECOG/WHO = Eastern Cooperative Oncology Group/World Health Organization, BMI = body mass index, ULN = upper limit of normal, LLN = lower limit of normal, AST = aspartate aminotransferase, ALT = alanine aminotransferase. Bold: Significant.

## Data Availability

The data presented in this study are available on request from the corresponding author. The data are not publicly available due to confidentiality rules of the original phase I clinical trials.

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
