# Peer review of "Clinical Predictors of Early Trial Discontinuation for Patients Participating in Phase I Clinical Trials in Oncology"

_cancers, 2021, doi:10.3390/cancers13102304_

Round 1

Reviewer 1 Report

Summary

In their article entitled “Clinical predictors of early trial discontinuation for patients participating in phase I clinical trials in oncology”, Douma et al. investigated which clinical predictors at baseline were significantly associated with early trial discontinuation of patients with cancer who participated in Phase I clinical trials. Data from 415 patients with solid tumors were pooled for the multilevel logistic regression analyses. Results identified 4 relevant predictors: 1) hyponatremia below normal limit; 2) elevated alkaline phosphatase level (>2.5 times the upper normal limit); 3) performance score of 1 or higher and; 4) opioid use. Based on the fact that hyponatremia was the strongest predictor, the authors propose to use this variable as a new eligibility criteria in order to reduce early trial discontinuation. Overall, this is a well-written study. The main objective remains highly relevant to the pursuit of early-phase clinical trials and addresses important safety and economic considerations. Statistical analyses are adequate. However, the following questions / comments should be addressed before publication is considered:

Questions / comments for the authors

  1. The abstract mentions that 67 trials were used in the analysis, however the results section reports only 66 (c.f. pages 1 and 3).
  2. The definition for early discontinuation included: 1) trial discontinuation within 28 days after the start of treatment and; 2) discontinuation before the administration of the first dose in eligible patients. How do these criteria compare to previously published literature on this topic?
  3. Did the authors consider the following variables for their analysis: 1) academic hospital (i.e., UMC vs John Hopkins) where the trials were conducted; 2) disease site; 3) type of therapy (immunotherapy vs others); 4) Phase I vs Phase I / II trials and; 4) and number of prior treatment regimens before enrollment?
  4. The authors mention that previous literature already identified performance status and elevated alkaline phosphatase levels as early discontinuation predictors. How can the authors explain the fact that their strongest predictor, hyponatremia, was never identified before in similar literature?
  5. In the discussion section, the authors mention a few explanations why patients with hyponatremia may do worse (e.g., SIADH, etc.): did any specific clinical features correlate with hyponatremia in the current study?
  6. The authors suggest to use hyponatremia as a potential eligibility criteria for future Phase I trials. Considering that hyponatremia may be associated with more advanced cancer (as stated by the authors in the discussion section), would this introduce a bias towards the inclusion of patients with lower disease burden? On the other hand, using this criteria may avoid selecting patients who do not have at least a 3-month life expectancy. What was the correlation of hyponatremia with 90-day mortality and OS in this analysis?

Author Response

  1. The abstract mentions that 67 trials were used in the analysis, however the results section reports only 66 (c.f. pages 1 and 3).
    • We would like to thank the reviewer for pointing out this error. We have now corrected this to 66, which is in concordance with the presented trials in the supplementary table.

“Data from 415 patients with solid tumors who participated in 66 trials were pooled for the current analysis” (page 1, lines 25-26).

  1. The definition for early discontinuation included: 1) trial discontinuation within 28 days after the start of treatment and; 2) discontinuation before the administration of the first dose in eligible patients. How do these criteria compare to previously published literature on this topic?
  • We would like to thank the reviewer for this relevant question. Two earlier studies have investigated early trial discontinuation in patients who participated in phase I trials. One study used trial discontinuation within 21 days and the other study used the inability to begin cycle 2 of therapy as the definition of early trial discontinuation. These definitions slightly differ from the definition of trial discontinuation within 28 days after start of treatment, that we have used in our study. We have chosen to add discontinuation before the administration of the first dosage in eligible patients to our definition of early trial discontinuation, because we are convinced that this is also a clinically relevant definition. Eligible patients who discontinue the trial before the administration of the first dose are not yet exposed to possible toxic treatments, but have already completed an intensive screening process with potential harmful interventions, such as biopsies. We have added a clarifying sentence to the method section.

Patients who met these criteria have not had a chance to benefit from the treatment, although they may have experienced toxicity or complications from treatment or investigations during the screening process.” (page 2, line 93-95).

  1. Did the authors consider the following variables for their analysis: 1) academic hospital (i.e., UMC vs John Hopkins) where the trials were conducted; 2) disease site; 3) type of therapy (immunotherapy vs others); 4) Phase I vs Phase I / II trials and; 4) and number of prior treatment regimens before enrollment?
    • We thank the reviewer for this useful comment. We have considered all above mentioned variables for our analyses. However, we aimed to identify clinical or patient-related predictors, which could be used as additional eligibility criteria across trials in the future. Therefore, we have not included the site of the trial, tumor type, type of therapy and phase of the trial as predictors in our analysis. However, we have checked whether the site of the trial (Amsterdam UMC versus Johns Hopkins), type of therapy (immunotherapy versus others) and phase of the trial (phase I versus phase I/II) were associated with early trial discontinuation, but none of the associations were statistically significant. Due to the large number of different types of primary tumors we were unable to perform a reliable analysis. However, the multilevel analysis which we performed to take the clustering of patients in trials into account, also partly corrects for tumor type, because many trials were tumor specific. The number of prior treatment regimens before enrollment could have been used for our analysis, but unfortunately data regarding the prior treatments were not available for our analysis. We have added this information to the introduction and method section:

“We here investigated clinical or patient-related predictors of early trial discontinuation of participants of various types of phase I trials, which could be used as additional eligibility criteria across trials in the future.” (page 2, lines 71-73)

“We also checked whether the hospital where the trial was performed (Amsterdam UMC, Vrije Universiteit Amsterdam versus The Sidney Kimmel Comprehensive Cancer Center at Johns Hopkins), type of therapy (immunotherapy versus other types of treatment) or type of the trial (phase I versus phase I/II) were non-patient related predictors of early trial discontinuation, but none of the associations were statistically significant.” (page 3, line 126-131)

  1. The authors mention that previous literature already identified performance status and elevated alkaline phosphatase levels as early discontinuation predictors. How can the authors explain the fact that their strongest predictor, hyponatremia, was never identified before in similar literature?
    • We would like to thank the reviewer for this question. We cannot explain why hyponatremia has not been investigated before as a predictor for early trial discontinuation. It has however been identified as a significant predictor for 90-day mortality and overall survival. This is acknowledged in the discussion section (page 6, lines 192-196).
  1. In the discussion section, the authors mention a few explanations why patients with hyponatremia may do worse (e.g., SIADH, etc.): did any specific clinical features correlate with hyponatremia in the current study?
    • We would like to thank the reviewers for this relevant suggestion. We have reviewed the cases of hyponatremia extensively and tried to identify patterns or causes of hyponatremia. However, we were not able to identify specific clinical features (e.g. comorbidity, use of painkillers, co-medication) which were correlated with hyponatremia. We have added an extra sentence to the methods section to clarify this.

We could not identify any corresponding clinical etiology (e.g. comorbidity, use of painkillers, co-medication) for the hyponatremia in the patients from our trials. Unfortunately, extensive diagnostics (e.g. MRI-scan of the brain) to determine the cause of the hyponatremia were not performed in most patients. However, hyponatremia may also reflect a more advanced, otherwise undetected, stage of cancer in general.” (page 6, lines 203-208)

  1. The authors suggest to use hyponatremia as a potential eligibility criteria for future Phase I trials. Considering that hyponatremia may be associated with more advanced cancer (as stated by the authors in the discussion section), would this introduce a bias towards the inclusion of patients with lower disease burden? On the other hand, using this criteria may avoid selecting patients who do not have at least a 3-month life expectancy. What was the correlation of hyponatremia with 90-day mortality and OS in this analysis?
    • We would like to thank the reviewer for raising this important question. It is possible that using hyponatremia as an exclusion criterion might cause the inclusion of patients with a lower disease burden. However, more importantly it will result in the exclusion of patients with the most advanced cancer, who are unlikely to benefit from toxic treatment and who are at risk for early trial discontinuation. This is supported by the finding that hyponatremia is also significantly associated with 90-day mortality (OR=3.62, 95% CI=1.63-8.02). The association between hyponatremia and 90-day mortality was also found in earlier studies. Since our study focused on predictors of early trial discontinuation we did not present a full analysis of the predictors of 90-day mortality and we did not have data regarding overall survival. However, we have added a sentence in the discussion section to emphasize that hyponatremia is an important predictor of both early trial discontinuation and 90-day mortality. Furthermore, we fully agree that changes to the conduct of early phase trials should be carefully considered because they potentially have far-reaching effects, which is acknowledged in the discussion section. 

“In an ancillary analysis, we found that hyponatremia was also significantly associated with 90-day mortality (OR=3.62, 95%CI=1.63-8.02).” (page 6, lines 196-197)

Reviewer 2 Report

Very pertinent study. Limitations are acknowledged by the authors. I have following questions which may be clarified in the manuscript by the authors:

1) What was the rationale for this eligibility criteria: discontinuation before administration of the first dosage in patients who were found to be eligible after the screening process?

2) Did hyponatremia, which was a major predictor of early discontinuation, correlate with brain metastasis? Were baseline brain MRI result available in these patients? Same for elevated alkaline phosphatase. Did this correlate with bone metastasis or liver metastasis on the scans?

Author Response

  1. What was the rationale for this eligibility criteria: discontinuation before administration of the first dosage in patients who were found to be eligible after the screening process?
    • We would like to thank the reviewer for this useful and relevant question. We have specifically chosen to add discontinuation before the administration of the first dosage in eligible patients to our definition of early trial discontinuation, because we are convinced that this is a clinically relevant definition. Eligible patients who discontinue the trial before the administration of the first dosage are not yet exposed to possible toxic treatments, but have already completed an intensive screening process with potential harmful interventions, such as biopsies. We have added a clarifying sentence to the method section.

Patients who met these criteria have not had a chance to benefit from the treatment, although they may have experienced toxicity or complications from treatment or investigations during the screening process.” (page 2, line 93-95).

  1. Did hyponatremia, which was a major predictor of early discontinuation, correlate with brain metastasis? Were baseline brain MRI result available in these patients? Same for elevated alkaline phosphatase. Did this correlate with bone metastasis or liver metastasis on the scans?
    • We would like to thank the reviewer for this question. Unfortunately baseline MRI-scans of the brain were not required for every study and could therefore not be used. Also the exact locations of the metastatic sites were not regularly collected in all the phase I trials, so they could not be used for our analysis. On the other hand, we also included levels of AST and ALT as potential markers for liver metastases in our analysis. An elevated AST was significantly correlated with early trial discontinuation in the univariable analysis, but not in the multivariable analysis. So this could mean that the significant correlation between an elevated alkaline phosphatase and early trial discontinuation is not only related to liver metastases (but also bone metastasis) or that alkaline phosphatase is a more sensitive marker for liver metastasis. However, we don’t have available data to investigate the underlying mechanism any further, which we have now clarified in the discussion section:

“We could not identify any corresponding clinical etiology (e.g. comorbidity, use of pain-killers, co-medication) for the hyponatremia in the patients from our trials. Unfortunately, extensive diagnostics (e.g. MRI-scan of the brain) to determine the cause of the hyponatremia were not performed in most patients. However, hyponatremia may also reflect a more advanced, otherwise undetected, stage of cancer in general.” (page 6, lines 203-208)

Round 2

Reviewer 1 Report

Thank you very much for addressing the review questions. Of note: the p values could be added to the new text on page 3, lines 126-131.